# Net Energy Value of a Cassava Chip Ration for Lactation in Holstein–Friesian Crossbred Dairy Cattle Estimated by Indirect Calorimetry

**DOI:** 10.3390/ani13142296

**Published:** 2023-07-13

**Authors:** Thidarat Gunha, Kanokwan Kongphitee, Bhoowadol Binsulong, Kritapon Sommart

**Affiliations:** 1Department of Animal Science, Faculty of Agriculture, Khon Kaen University, Khon Kaen 40002, Thailand; thidarat.gunha@kkumail.com (T.G.); b.bhoowadol@kkumail.com (B.B.); 2Renewable Energy and Environmental Engineering Program, Institute of Interdisciplinary Studies, Rajamangala University of Technology Isan, Nakhon Ratchasima 30000, Thailand; kanokwan.aof1@gmail.com

**Keywords:** tapioca, digestibility, methane, dairy cow, calorimetry

## Abstract

**Simple Summary:**

The net energy value of feedstuff for lactation is a critical piece of data for the development of the dairy feeding system toward improving animal productivity and sustainability. Cassava (*Manihot esculenta* Crantz), a tuber crop in the tropics, is available year-round for Thai cassava starch and the export feed industry. It contains abundant digestible starch, so its nutritive value makes it suitable as an alternative energy feed source for ruminants. Determination of the net energy value of cassava chips for lactation will allow for the more precise formulation of diets that are optimal for dairy cows. This study aimed to (1) evaluate the effects of cassava, including in the diets offered to mid-lactation dairy cows, on production performance and (2) determine the net energy for lactation content of cassava chips. The findings indicate that increasing the cassava chips in the diets of Holstein–Friesian crossbred dairy cows could result in improved nutrient intake, digestibility, energy balance status, milk yield, and milk composition without the effect on enteric methane yield and intensity. The net energy for the lactation of cassava chips was estimated as 8.03 MJ/kg DM.

**Abstract:**

The objectives of this research were to (1) determine the feed intake, digestibility, and energy utilization and (2) estimate the net energy value of cassava chips consumed by lactating dairy cows. Four multiparous Holstein–Friesian crossbred cows at 139 ± 33 (mean ± SD) day in milk were assigned according to a 4 × 4 Latin square design with four periods. The four treatments included a diet substituted with cassava chips on a 0%, 12%, 24%, and 36% dry matter (DM) basis in the basal diet. Indirect calorimetry with a head cage respiration system was used to determine nutrient and energy utilization. Increasing the number of cassava chips in the diet resulted in a linear increase (*p* < 0.05) in nutrient intake and digestibility but a linear decrease (*p* < 0.01) in crude protein (CP) and fiber. The enteric methane yield and intensity were not affected (*p* > 0.05), while energy was lost as feces and urine reduced linearly (*p* < 0.05). Milk yield and milk composition (protein, fat, lactose) also increased linearly (*p* < 0.05). The net energy requirement for the maintenance of the lactating cows was estimated as 327 kJ/kg of metabolic body weight, and the efficiency of metabolizable energy used for lactation was 0.66. The estimated net energy value of cassava chips for lactation was 8.03 MJ/kg DM.

## 1. Introduction

Cassava (*Manihot esculenta* Crantz) is a tuber crop that is widely grown in tropical and subtropical zones. In Thailand, cassava is an economically important crop, with around 32.96 million tons produced in 2021 [1], and approximately 54% of cassava root production is used for starch, 44% for animal feed and exports, and 2% for ethanol production [2,3,4]. Cassava chips are a processed form of fresh cassava root that have been washed, chopped into small pieces, and sun-dried for 3 to 6 days on concrete floors, depending on weather and sunshine. Cassava chips are used in the diets of cattle (beef and dairy) and goats at substitution levels of 25 to 75% [3,5]. Cassava chips have high non-fiber carbohydrate (69 to 82%) [3,6,7,8] but low crude protein (2 to 3% of CP) contents [3,4], resulting in a faster rate of rumen degradability than other energy sources in ruminant feeds [9], and they improve rumen fermentation and digestibility [5]. Sommart et al. [3] also reported that increasing the proportion of cassava chips up to 30% in the total mixed ration of the lactating dairy cow diet significantly improves nutrient intake, milk yield, and milk composition. Cassava chips have great potential to be used as an energy feed source for dairy cattle production, especially in ensuring the quantity of produce and its year-round availability in Thailand. However, in the research on cassava chips as lactating dairy cow feed, information on the energy utilization of lactating cows is lacking. Determination of the energy value of diets containing cassava chips will allow for the more precise formulation of diets for dairy cows.

The objectives of this study are to (1) investigate the effect of including cassava chips on feed intake, digestibility, energy utilization, and milk production and (2) determine the net energy value of cassava chips for lactation in Holstein–Friesian crossbred dairy cows using animal calorimetry techniques.

## 2. Materials and Methods

This research was conducted at the University farm research station, Khon Kaen, Thailand (latitude 16.48° N and longitude 102.82° E), between December 2020 and March 2021. All animal-related procedures were reviewed and approved by the Ethics of Animal Experimentation of the National Research Council of Thailand (Record No. IACC-KKU-10/63, Reference No. 660201.2.11/17).

### 2.1. Animal, Experimental Design, and Diets

Four multiparous Holstein–Friesian (HF) crossbred lactating cows (88.6 % HF, 11.4% native Thai cattle), non-pregnant, with mean initial body weight (mean ± SD) 483.3 ± 10.0 kg, milk production of 14.7 ± 5.6 kg/d, and 139.0 ± 33.2 days in milk, were used in this study. Cows were milked in a pen twice daily at 06:30 am and 3:30 pm and fed fermented total mixed rations ad libitum at 08:00 and 16:00 h.

The experimental design was a 4 × 4 Latin square design, in which each cow was randomly assigned to 1 of 4 dietary treatments and alternated over four periods. Each experimental period lasted 21 days, beginning with 14 days of adaptation, and followed by daily measurements of milk, feces, urine, and respiratory gas exchange from day 17 to day 21 (5 days). Throughout the experiment, cows were housed individually in pens (2.5 m × 4.5 m), with free access to feed and clean drinking water.

The four dietary treatments included one basal diet (0% DM of cassava chip) and three test diets. For the three test diets, rice straw, cassava pulp, wet brewery waste, rice bran, palm kernel cake, and soybean meal in the basal diet were substituted by cassava chips at the ratios of 12%, 24%, and 36% DM, respectively (Table 1). The fermented total mixed ration was produced monthly using a horizontal mixer; approximately 2000 kg of dietary treatment ingredients were mixed and then packed into 35 kg per bag and ensiled into high-density polyethylene bags, according to Kongphitee et al. [10]. The substitution method was used to determine the net energy value of the cassava ingredient according to Wei et al. [11]. 

### 2.2. Feed Intake and Digestibility

The dairy cows were weighed and recorded at 07:30 on the first and last day of each experimental period to determine their dry matter intake as a percentage of body weight and metabolic body weight. Throughout the feeding period, the individual daily feed intake was recorded as the difference between the amount of feed offered and the feed refused.

The total collection technique was conducted in an indoor digestion trial pen (165 × 375 cm) installed with a head cage respiratory gas system (width 105 cm × depth 80 cm × height 173 cm) for each animal over 5 consecutive days. The animals were relocated to a digestion trial pen on days 17 to 21 of each experimental period. Feces and urine amounts were measured daily. The excreted feces were collected using pans placed behind the animal. A tube urine cup collected the total urine volume in plastic tanks. The total urine volume was collected in polypropylene bags containing 6 N hydrochloric acid to maintain the pH below 3.0. Feces (1 kg) and urine (120 mL) were sampled daily and stored at 4 °C. After the collection period was completed, 1 kg daily aliquot samples of offered feed, refusals, and excreted feces and 120 mL of excreted urine were well-mixed and then stored at −18 °C.

### 2.3. Animal Calorimetry

During the digestion trial, the oxygen consumption, carbon dioxide, and methane production of each animal were measured using a head cage respiration chamber system, according to Suzuki et al. [12]. Briefly, the system consisted of a head cage, a gas sampling and analysis unit, and a data acquisition and processing unit. Cows kept their heads in the hood and had access to feed trays and automatic water throughout the day, as well as the ability to lie down on the rubber-mat floor. A head hood and a flow meter (NFHY-R-O-U, Nippon Flow Cell, Tokyo, Japan) were used to measure the flow rate and total volume of air flowing out of the respiration chamber. The oxygen, carbon dioxide, and methane concentration in the inflow and outflow lines was measured using a gas analyzer (4100 Gas Purity Analyzer, Servomex Group, East Sussex, UK). The gas analyzers were calibrated daily using the certified gases (Takachiho Chemical Industrial Co., Tokyo, Japan), including two oxygen concentrations (19.0 and 20.6%), 1.89% of carbon dioxide, and 1960 ppm of methane. Calorimetric system recovery gas tests (98 to 104%) were conducted using the carbon dioxide injection method. The respiratory gas exchange measurements were taken at intervals of 7.5 min for 23.30 h per day from the initial day at 08:00 am to 07:30 am the next day to determine energy partition and consumption over the last three days of the respiration collection period. Heat production was calculated according to the Brouwer method [13] (Equation (4)), and the energy partition was estimated [12].

### 2.4. Sample Collection and Chemical Analysis

Feed samples were collected during each period to determine the nutritive value. The dry matter of feeds and feces was analyzed in a fan-forced oven at 105 °C. Each subsample (800 g fresh matter) was dried in a fan-forced air oven at 60 °C for 72 h and then ground (1 mm screen size). Afterward, feeds and feces were analyzed using the Association of Official Analytical Chemists (AOAC) methods [14] for DM, ash, CP, and ether extraction (EE) (method numbers 967.03, 942.05, 984.13, and 920.39, respectively). The neutral detergent fiber (NDF) and acid detergent fiber (ADF) content was analyzed using a fiber analyzer (ANKOM 200/220, ANKOM Technology, Macedon, NY, USA), NDF treated with thermostable alpha-amylase, and sodium sulfite [15,16]. Non-fiber carbohydrate (NFC) content was calculated according to the equation NFC (%) = 100 − (%CP + %NDF + %EE + %Ash). The gross energy content was determined using a bomb calorimeter (IKA C2000 Basic, IKA-Werke, Staufen, Germany).

The milk yield was recorded daily during the morning (06:30 h) and afternoon (15:30 h) milking, and milk samples were also collected for 5 consecutive days. One aliquot (110 mL) was refrigerated at 4 °C and sent to the Veterinary Research and Development Center, Upper Northeastern Region (Khon Kaen, Thailand) to determine the milk fat, protein, solid non-fat (SNF), and lactose concentrations using MilkoScan™ 7RM (Foss Electric, Denmark) and somatic cell count using the Milk Analyzer Fossomatic™ 7 DC (Fossomatic DC, Denmark). The second aliquot (110 mL) was frozen at −20 °C; then, gross energy was analyzed using a bomb calorimeter (IKA C2000 Basic, IKA-Werke, Staufen, Germany).

### 2.5. Calculation

Fat and protein-corrected milk (FPCM, kg/d) was calculated according to Gerber et al. [17], as in Equation (1):FPCM = (0.337 + 0.116 × fat% + 0.06 × protein%) × milk yield (kg/d)(1)

Milk energy (MJ/kg) was calculated according to Cabezas-Garcia et al. [18] units for fat, protein, and lactose in milk are g/kg, as in Equation (3): Milk energy = (0.0384 × fat) + (0.0223 × protein) + (0.0199 × lactose) − 0.108(2)

Energy-corrected milk (ECM, kg/d) was calculated according to Cabezas-Garcia et al. [18], as in Equation (3):ECM = (milk yield (kg/d) × milk energy (MJ/kg))/3.1(3)

Heat production (HP, kJ/d) was calculated according to Brouwer [13], as in Equation (4)
HP = 16.18 × O_2_ + 5.02 × CO_2_ − 5.99 × UN − 2.17 × CH_4_,(4)
using volumes of O_2_ consumption (L/d), CO_2_ production (L/day), CH_4_ production (L/d), and urinary nitrogen excretion (g/d). Methane energy (kJ/d) was calculated according to Blaxter and Clapperton [19] as CH_4_ = 39.54 kJ/L × CH_4_ (L/d).

Energy balance (EB; kJ/kg BW^0.75^) was calculated using Equation (5).
EB = ME intake − HP − Milk energy(5)

The efficiency of ME use for lactation (*k*_l_) was calculated according to Moe et al. [20] using Equation (6).
*k*_l_ = E_l(0)_/(ME intake − ME_m_)(6)
where E_l(0)_ is the milk energy output (E_l_) adjusted to zero energy balance (kJ/d) and calculated from Equations (7) and (8). ME intake is calculated as follows: ME intake = gross energy intake − fecal energy − urinary energy − methane energy. The ME_m_ is the ME requirement for maintenance (kJ/d).
EB > 0, E_l(0)_ = E_l_ + 0.95 × EB(7)
EB < 0, E_l(0)_ = E_l_ − 0.84 × EB(8)

The substitution and multiple substitution regression methods were used to determine the metabolizable and net energy value of the test ingredient according to Wei et al. [11]. The DE, ME, and net energy for lactation (NE_L_) values of test ingredients were calculated by the substitution method, as described by Wei et al. [11] in Equation (9).
E_ti_ (MJ/kg DM) = (E_td_ − (1 − P_ti_) × E_bd_)/P_ti_(9)
where E_ti_ is the energy value of the test ingredient, E_td_ (MJ/kg DM) is the energy value of the test diet, E_bd_ (MJ/kg DM) is the energy value of the basal diet, and P_ti_ is the test ingredient substitution ratio in the basal diet.

Alternatively, the DE, ME, and NE_L_ values in test ingredients can be estimated by the regression of the test ingredient-associated energy intake (MJ/d) against kilograms of the test ingredient substitution amount intake (kg/d), with the slope of regression equation representing the energy concentration in MJ/kg DM. The product of E_ti_ at each level of test ingredient cassava chips substitution rate energy concentration (MJ/kg) and kilograms of cassava chips intake (0.12, 0.24, or 0.36 kg) is the cassava chips-associated DE, ME, or NE_L_ intake in kilocalories [11].

### 2.6. Statistical Analysis

Regression equations to estimate the energy value of cassava were generated using the REG procedure of SAS [21]. The dependent variables in the prediction equation were casava-associated energy intake, respectively, and the independent variable was test ingredients intake.

The experimental data were analyzed using the general linear model (GLM) of SAS for a 4 × 4 Latin square design using the following statistical model:Y_ijk_ = μ + ρ_i_ + γ_j_ + τ_k_+ ε_ijk_
where Y_ijk_ is a dependent variable; μ is the mean for all observations; ρ_i_ is the fixed effect of the period (i = 1 to 4); γ_j_ is the fixed effect of the cow (j = 1 to 4); τ_k_ is the fixed effect of dietary treatment (k = 1 to 4), and ε_ijk_ is the residual error. Within treatments, linear, quadratic, and cubic contrasts were evaluated to determine the effect of increasing proportions of cassava chips in the diet. 

## 3. Results

### 3.1. The Chemical Composition and Energy Content

The chemical composition, energy content of cassava chips, and diets are presented in Table 2. The OM, NDF, and ADF contents in dietary treatments were similar among diets. The CP and EE were decreased, but NFC and ME content increased substantially when increasing cassava in the diets.

### 3.2. Feed Intake and Digestibility

Increasing the substitution of cassava chips in the diets resulted (Table 3) in a linear increase (*p* < 0.01) in DM intake, organic matter (OM) intake, and NFC intake, whereas there was a cubic increase (*p* < 0.01) in CP, EE, NDF and ADF intake.

Apparent nutrient digestibility is shown in Table 4. It was clearly observed that as cassava chip substitution in the basal diet increased from 0 to 36% DM in the dietary treatment, the apparent digestibility of DM, OM, and NFC increased linearly (*p* < 0.05), whereas the digestibility of CP decreased linearly (*p* = 0.02). The digestibility of EE, NDF, and ADF was not affected by dietary treatment.

### 3.3. Milk Yield and Composition

The substitution of cassava chips in the dietary treatment did not affect the milk yield (kg/d) (Table 5), but there were linear increases in fat and protein-corrected milk yield (FPCM), energy-corrected milk yield, and the yield of milk protein and fat (*p* < 0.05). The composition of milk protein, fat, lactose, solid non-fat, and milk energy increased linearly (*p* < 0.05) with the increasing cassava chip level of the diet (Table 5). The somatic cell count was not affected by the proportion of cassava chips (*p* > 0.05). The feed efficiency was unaffected by the dietary treatments (*p* > 0.05).

### 3.4. Respiratory Gas Consumption and Production

Results of oxygen consumption, carbon dioxide production, and enteric methane emission analysis in lactating dairy cows fed cassava chips substituted into the basal diet are presented in Table 6. The carbon dioxide production and respiratory quotient increased linearly (*p* < 0.01) with the increasing level of cassava chip substitution in the diet.

Methane yield (L/kg DMI) and methane intensity (L/kg FPCM) were not affected (*p* > 0.05) by dietary treatment. Enteric methane emission (L/d) increased linearly (*p* < 0.05) as cassava chip substitution was increased in the basal diet.

### 3.5. Energy Partitioning

The results regarding the effects of substituting cassava chips in the basal diet on energy intake and utilization, as determined by metabolic body size, are shown in Table 7. Heat production, milk energy, and energy balance were unaffected by the proportion of cassava chips in the diet (*p* > 0.05). Energy lost as feces and urine decreased linearly (*p* < 0.05), but there were tendencies for linear increases in methane energy lost (*p* = 0.06). 

Intake of DE tended to increase (*p* = 0.06), and ME increased linearly (*p* < 0.05). The DE and ME content of the diets increased linearly (*p* < 0.05). The efficiency of energy utilization (DE/GE, ME/GE, and ME/DE ratios) increased linearly (*p* < 0.05) with the increasing level of cassava chip substitution in the diet.

### 3.6. Energy Requirement for Maintenance and Efficiency of ME Utilization for Lactation 

Estimations of the maintenance energy requirements for lactating dairy cows, determined by linear regressions of milk energy output adjusted to zero energy balance (E_l(0)_, kJ/kg BW^0.75^) against ME intake (kJ/kg BW^0.75^), are presented in Figure 1. Regression (E_l(0)_ = 0.59 × ME intake—310.7) was highly significant (*p* < 0.01), and the *R^2^* value was 0.66. The metabolizable energy requirement for the maintenance (ME_m_) of Holstein–Friesian crossbred dairy cows in this experiment was 527 kJ/kg BW^0.75^, and the efficiency of ME utilization for lactation was 59%. The net energy requirement for maintenance (NE_m_), as derived, was 310.7 kJ/kg BW^0.75^.

### 3.7. Energy Values of Cassava Chips for Lactation by Substitution and Regression Methods

The regression of cassava chip-associated energy intake (*Y*, MJ/d) against the cassava chip substitution amount (*X*, kg/d) is shown in Table 8. The regression equation for the DE was *Y* = 12.42*X* + 0.002 (R^2^ = 0.99, RMSE = 0.17, *p* < 0.001, n = 16); the ME was *Y* = 10.63*X* + 0.002 (R*^2^* = 0.98, RMSE = 0.20, *p* < 0.001 n = 16), and it was determined that the cassava chips had energy value for lactation of 12.42 MJ DE/kg DM and 10.63 MJ ME/kg DM. In addition, the NE_L_ regression equation was *Y* = 8.04*X* + 0 (R^2^ = 0.76, RMSE = 0.65, *p* < 0.001, n = 16), and it was determined that the energy value for cassava chips was 8.04 MJ NE_L_/kg DM.

The DE, ME, and NE_L_ values of cassava chips derived based on a single substitution ratio were similar between the substitution and regression methods (Table 9). Energy values of cassava chips for lactation, estimated by the substitution method, were 12.43 MJ DM/kg DM, 10.64 MJ ME/kg DM, and 8.03 MJ NE_L_ /kg DM, respectively.

## 4. Discussion

### 4.1. Nutrient Intake, Total Tract Digestibility, and Milk Production

Feed intake is a limiting factor determining nutrient digestibility and energy supply for maintenance and production. Tropical feeds and feeding systems often depend on low-quality feed sources that limit nutrient digestibility and energy intake. This study examined the effects of including cassava chips in the diet of Holstein–Friesian crossbred dairy cows on feed intake, digestibility, energy utilization, and milk production. When cassava chips were replaced in the basal diets, overall feed intake, digestibility, energy intake, and negative energy balance status, and, thus, improvements in milk production.

In the present study, dry matter, organic matter, non-fiber carbohydrate intake, and digestibility all exhibited substantial improvement, suggesting that a greater increase in nutrients and energy supply to lactating dairy cows was achieved with increased cassava chip inclusion in the diet. These results are consistent with previous observations [3,4,5,10], which indicated that cows fed higher dietary NFC content in silage had a significantly greater energy supply and, thus, a greater positive energy balance. The DM, OM, and NFC intakes increased when increasing the level of cassava chips in the diet, which resulted in increased digestibility of DM, OM, and NFC without any adverse effect on fiber digestion and rumen fermentation. The decreased intake of CP and EE and fiber in dairy cows fed cassava chips at 12, 24, and 36%, respectively, might have been due to the lower CP and EE content of cassava chips compared to the basal diets. Similarly to the present study, Sommart et al. [3] observed a decrease in CP intake of lactating dairy cows with increasing cassava chip content, replacing corn grain at a proportion of 13.5 to 54.0% in the concentrate.

The highest milk production response is associated with the highest feed intake and digestibility of ruminal and intestinal starch, which supplies glucogenic precursors through rumen fermentation end products and exogenous glucose absorption. Our data indicate that lactating dairy cows fed up to 36% of cassava chips in the diet have greater non-fiber carbohydrate digestibility and metabolizable energy supply, thus leading to an improved milk yield and milk composition. Paengkoun and Paengkoun [22] similarly reported that the addition of up to 45% of cassava chips in the concentrate of dairy cows improved milk fat and milk protein yields. Moreover, a similar report by Sommart et al. [3] indicated that the milk fat and milk protein concentrations increased progressively with increasing levels of cassava in lactating cows [3,4,22].

### 4.2. Enteric Methane Emissions

Enteric methane is a natural byproduct of the rumen microbial fermentation of carbohydrates and, to a slightly lesser degree, amino acids [23]. In the present study, a significant linear increase in daily enteric methane emissions (L/d) but no effect on the methane yield (L/kg DMI) and intensity (L/kg PFCM) indicates that the total carbohydrate fermentation and digestibility improved; thus, the efficiency of energy utilization increases when substituting cassava chips in the basal diet. The OM substrates are fermented in the rumen, producing short-chain volatile fatty acids, carbon dioxide, and metabolic hydrogens, which hydrogen-combines with carbon dioxide by rumen methanogen synthesis to produce methane [24,25]. In this study, daily methane emissions were similar to Mickayla’s [25], who reported that methane emissions of lactating dairy cows ranged from 329 to 360 L/d.

The enteric methane conversion factor is used to assess the methane estimation and global warming impact of the national inventories [26]. In our study results, the enteric methane conversion factor ranged from 6.4 to 7.7%, which was higher than the IPCC 6.5% default value for cattle in developing countries [26]. The result is also in good agreement with the results of Chuntrakort et al. [27]. Kurihara et al. [28] found that mature Brahman cattle fed tropical forage-based diets had enteric methane conversion factors ranging from 6.7 to 11.4%. Our results are also consistent with previous work [10,12,19,27,29] reporting a high rate of enteric methane energy factor of zebu cattle fed low-quality roughage-based diets. The typically high lignocellulose content of feed available in tropical dairy cattle production systems may be the main factor affecting methane emissions. Improving feed quality can be an important strategy to reduce enteric methane emissions. Methane emissions are influenced by feed intake [29], forage quality and species, carbohydrate type, physical processing, forage preservation, and feeding frequency [24].

The respiratory quotient (RQ) is the ratio of carbon dioxide production to oxygen consumption [30]. The RQ value greater than 1 can be used as an approximate indicator of carbohydrates being utilized for fat synthesis that is occurring in the animal [31]. The RQ values increased when increasing the level of cassava chips in the diet; the average value of RQ in this study (ranging from 1.13 to 1.19) was close to the RQ values reported for dairy cows using the respiration chamber technique, which ranged from 1.01 to 1.17 [30,32]. Our result indicated that NFC intake and digestibility increase associated with the increasing RQ, milk fat content, and yield, confirming a higher milk fatty acid synthesis from carbohydrate utilization when increasing the level of cassava chips in the diet.

### 4.3. Energy Partition, Efficiency of Utilization, and Maintenance Requirement of Dairy Cows

In this study, increasing the proportion of cassava chips in the diet significantly improved the energy intake and energy balance of Holstein–Friesian crossbreeds. Increases in DE and ME intake were related positively to decreases in energy loss from feces and urine; therefore, a greater positive energy balance was achieved. The reduction in energy loss by feces and urine was positively related to potential increases in DE and ME intake as the energy content of the diet increased, which was similar to the results reported by Kongphitee et al. [10]. We also observed that the dairy cows with high ME intake had improved energy efficiency and increases in the ratios of DE to GE, ME to GE, and ME to DE. Tyrrell and Moe [33] reported that the ratio of ME to DE increases as intake improves (i.e., partial compensation for digestibility depression due to decreased feces and urine energy loss), resulting in a less negative intercept when ME is compared to DE as a measure of intake.

Estimation of maintenance energy requirements was achieved through regression of the milk energy output adjusted to zero energy retention against the ME intake (Figure 1). The analysis of the pooled data determined the ME_m_ requirement of 527 kJ/kg BW^0.75^ for Holstein–Friesian crossbred dairy cows. Our result is in good agreement with the value of 519 kJ/kg BW^0.75^ recently reported by Olivera et al. [34] when using a linear mixed regression of metabolizable energy intake and milk energy output adjusted to zero energy balance, derived from data from 13 studies, involving 49 treatment means for *Bos taurus × Bos indicus* crossbred cows under tropical conditions using no grazing dataset. However, the ME_m_ in this study was less than the value of 610 kJ/kg BW^0.75^ reported by Morris and Kononoff [35] for Jersey cows and 688 kJ/kg BW^0.75^ reported in Dong et al. [36] when using linear regression of milk energy output adjusted to zero energy balance (E_l(0)_) against ME intake for lactating Holstein–Friesian cows (*n* = 823), and it was also lower than the values suggested for Holstein and Jersey–Holstein crossbred dairy cows (710 and 670 kJ/kg BW^0.75^, respectively) by Xue et al. [37] using data derived from a repeated factorial design study with eight Holstein and eight Jersey–Holstein crossbred dairy cows. The variability in energy requirements for maintenance was primarily attributed to the animal species, state, and feeding system [11,34,35,36,38,39].

In the present study, the mean *k*_l_ value was 0.59, which was almost identical to the reported value of 0.53 of *Bos taurus* × *Bos indicus* dairy cows under tropical conditions by Oliveira [34] but lower than the value of *Bos taurus* [31,36,39]. Oliveira [34] reported that the lower *k*_l_ value of the crossbred dairy cows in tropical conditions could be due to the genotypes, diets, and differences in homeostatic regulation. These discrepancies between the estimated and observed maintenance expenditures and efficiency of ME use for lactation reflect and highlight the progress of modern dairy cattle breeding, which focuses on improvements through genetic selection [35].

### 4.4. Estimation of Net Energy Value of Cassava Chips for Lactation by Substitution Compared with Regression Methods

With the energy estimated for individual feedstuffs, the substitution method was used to resolve the limitations of the direct method by feeding the test feedstuff in combination with a basal diet of known energy value [11]. In the present study, the ME values of cassava chips were not significantly different, whether calculated by the substitution method or the linear regression method (value 10.64 MJ ME/kg DM, Table 8). In addition, the NE_L_ value of cassava obtained by the substitution method (8.03 MJ NE_L_ /kg DM) was similar to that obtained with the linear regression method. Recently, the regression method has been used to evaluate the energy available to ruminants from a given feedstuff. Wei et al. [11] determined the ME and NE content of rice straw (6.76 and 3.42 MJ/kg DM, respectively) or wheat straw (6.43 and 3.28 MJ/kg DM, respectively) using the regression method as the substitution ratio ranged from 100 to 600 g/kg. These ME values, determined for cassava chips, are lower than those of 12.2 MJ ME/kg DM reported in WTSR [8] and 12.9 MJ ME/kg DM in Sommart et al. [3] but higher than 10.04 MJ ME/kg DM in WTSR [8]. In addition, the NE_L_ content (8.04 MJ NE_L_/kg DM) of cassava chips in this study was found to be 9.8% higher than the value of 7.32 MJ NE_L_/kg DM reported by WTSR [8]. The nutritive value varies according to the type of feedstuff, strain, harvesting process, weather, season, and evaluation method, resulting in different nutritional and energy values.

Although a 4 × 4 Latin square design accounted for cow and period effects and allowed for measurements of animal responses to changes in dietary treatment, the relatively short (21 d) time of the periods in the Latin square design and the limited number of cows per treatment are limitations to this study. In particular, the milk production results need to be confirmed over a longer feeding period and a greater number of animals. The limitation of this experiment was also due to the increase in the cassava levels in the diets by a single-point substitution or multiple-point substitution method required for evaluating the energy content of individual feedstuffs. The method involves different substitution rates of a test feed (cassava chips) into each energy feed source in the basal diets. Therefore, the effects observed are not only due to the cassava increase but also to diet composition change.

## 5. Conclusions

A study utilizing lactating dairy cows was conducted to determine the effects of increased inclusion of cassava on nutrient and energy utilization and the net energy value of cassava chips for lactation. The results indicate that when cassava chips are included at up to 36% in the diets of dairy cows, an increased feed intake, digestibility, energy intake, and energy balance, and, thus, improvements in milk production, are obtained. The NE_L_ value of cassava chips for Holstein–Friesian crossbred dairy cows obtained using the regression method by indirect calorimetry was 8.04 MJ NE_L_/kg DM, respectively. The energy value of cassava chips derived based on a single substitution ratio was similar between the substitution and regression methods. Long-term feeding trials or on-farm research is needed for the development of a practical and economical dairy zebu crossbred fed cassava-based diets feeding system.

## Figures and Tables

**Figure 1 animals-13-02296-f001:**
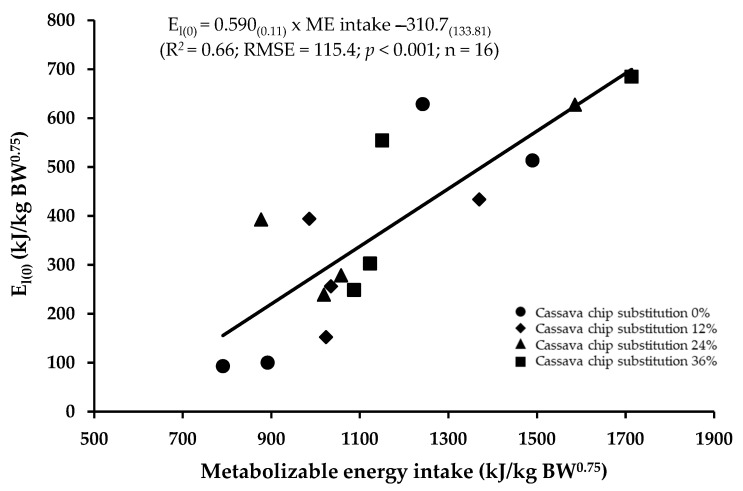
Linear regressions of milk energy output adjusted to zero energy balance (E_l(0)_, kJ/kg BW^0.75^) against ME intake (kJ/kg BW^0.75^) for lactating dairy cows fed fermented total mixed ration diet with different levels of cassava chip substitution.

**Table 1 animals-13-02296-t001:** Ingredients and feed costs of the four-cassava chip substitution dietary treatments.

Items	Cassava Chip Substitution (%)			
	Basal Diet (0)	12	24	36
Ingredients, %DM ^1^				
Rice straw	15.0	13.2	11.4	9.6
Cassava chips *	-	11.8	23.5	35.3
Cassava pulp	30.0	26.4	22.8	19.2
Wet brewery waste	15.0	13.2	11.4	9.6
Rice bran	11.0	9.7	8.4	7.0
Palm kernel cake	12.0	10.5	9.1	7.7
Soybean meal	15.0	13.2	11.4	9.6
Urea	0.5	0.5	0.5	0.5
Mineral ^2^	1.0	1.0	1.0	1.0
Premix ^3^	0.5	0.5	0.5	0.5
Total	100.0	100.0	100.0	100.0
Feed cost, TBH/kg FM	3.64	3.85	4.22	4.54

* Cost 6.88 TBH/kg FM. ^1^ DM = dry matter, FM = fresh matter. ^2^ Minerals included 93.72 g/kg Ca, 46.86 g/kg P, 107.78 g/kg Na, 18.56 g/kg S, 8.24 g/kg Mn, 7.49 g/kg Zn, 3.37 g/kg Mg, 1.17 g/kg Cu, 0.15 g/kg Co, 0.01 g/kg K, 0.04 g/kg I, and 0.02 g/kg Se (Mineral #0106410029, Dairy Farming Promotion Organization of Thailand (D.P.O.), Saraburi, Thailand). ^3^ Mineral and vitamin premix containing 5,000,000 IU/kg vitamin A, 1,000,000 IU/kg vitamin D3, 10,000 IU/kg vitamin E, 25 g/kg Fe, 4 g/kg Cu, 20 g/kg Mn, 0.13 g/kg Co, 15 g/kg Zn, 0.75 g/kg I, 0.38 g/kg Se, 0.20 g/kg of feed preservative, and 0.88 g/kg of feed additive (Golden Mix S#0104610040, DFC Advanced Co. Ltd., Khon Kaen, Thailand).

**Table 2 animals-13-02296-t002:** Analyzed chemical composition, energy content of cassava chips, and the four-cassava chip substitution dietary treatments.

Items ^1^	Cassava Chip	Cassava Chip Substitution (%)			
		0	12	24	36
Chemical composition, %DM					
Dry matter	87.4	31.6	33.3	37.3	40.1
Organic matter	97.7	92.9	93.3	93.3	93.9
Crude protein	2.1	18.5	16.4	13.9	13.8
Neutral detergent fiber	7.9	35.4	35.4	36.1	35.3
Acid detergent fiber	6.6	22.2	24.3	22.6	21.9
Ether extract	0.4	6.2	4.7	4.8	3.8
Non-fiber carbohydrate	86.9	32.8	36.8	38.4	40.4
Energy content					
Gross energy, MJ/kg DM	16.2	18.2	18.1	18.0	17.8
Metabolizable energy ^2^, MJ/kg DM	12.2	10.6	10.7	11.2	11.5

^1^ DM = dry matter. Non-fiber carbohydrate calculated as 100 − (%NDF + %CP + %EE + %ash). N = nitrogen. ^2^ Metabolizable energy of cassava chips calculated according to WTSR [8].

**Table 3 animals-13-02296-t003:** Daily feed and nutrient intake in lactating dairy cows (*n* = 4 per treatment) fed a fermented total mixed ration at different levels of cassava chip substitution.

Items ^1^	Cassava Chip Substitution (%)				SEM	*p*-Value ^2^		
	0	12	24	36		L	Q	C
Body weight, kg	480.9	486.4	493.9	491.8				
Dry matter intake								
kg/d	12.6	12.4	14.5	14.8	0.47	<0.01	0.56	0.10
%BW	2.5	2.6	3.0	3.1	0.13	0.03	0.65	0.18
Nutrient intake, kg/d								
Organic matter	11.8	11.6	13.5	13.9	0.44	<0.01	0.57	0.10
Crude protein	2.3	1.9	2.0	1.9	0.65	<0.01	0.03	0.06
Ether extract	0.8	0.6	0.7	0.6	0.02	<0.01	0.15	<0.01
Neutral detergent fiber	5.8	4.6	5.5	5.2	0.17	0.28	0.06	<0.01
Acid detergent fiber	3.0	2.2	3.2	2.9	0.09	0.12	0.06	<0.01
Non-fiber carbohydrate	2.9	4.5	5.4	6.3	0.18	<0.01	0.10	0.34

^1^ BW = body weight. BW^0.75^ = metabolic body weight. ^2^ Probability of significant linear (L), quadratic (Q), and cubic (C) effect of cassava chip levels.

**Table 4 animals-13-02296-t004:** Digestibility in lactating dairy cows (*n* = 4 per treatment) fed a fermented total mixed ration at different levels of cassava chip substitution.

Items	Cassava Chip Substitution (%)				SEM	*p*-Value ^1^		
	0	12	24	36		L	Q	C
Digestibility, g/kg DM								
Dry matter	653	676	703	712	13.8	0.02	0.62	0.74
Organic matter	685	713	741	750	11.4	<0.01	0.43	0.73
Crude protein	722	709	672	649	18.6	0.02	0.80	0.66
Ether extract	886	870	846	848	14.7	0.08	0.57	0.63
Neutral detergent fiber	482	570	540	603	45.4	0.15	0.79	0.34
Acid detergent fiber	383	507	513	535	49.4	0.08	0.34	0.57
Non-fiber carbohydrate	895	904	957	975	19.0	0.01	0.81	0.38

^1^ Probability of significant linear (L), quadratic (Q), and cubic (C) effects of cassava chip levels.

**Table 5 animals-13-02296-t005:** Milk production and chemical composition in lactating dairy cows (*n* = 4 per treatment) fed a fermented total mixed ration at different levels of cassava chip substitution.

Items ^1^	Cassava Chip Substitution (%)				SEM	*p*-Value ^2^		
	0	12	24	36		L	Q	C
Milk production, kg/d								
Milk yield	13.8	13.5	13.4	13.8	0.19	0.94	0.11	0.59
FPCM	13.5	13.9	13.9	15.3	0.50	0.05	0.32	0.45
ECM	13.6	14.1	14.1	15.8	0.60	0.05	0.31	0.44
Protein	440	440	431	489	12.8	0.05	0.06	0.23
Fat	539	578	585	668	34.7	0.04	0.55	0.52
Milk composition, g/kg								
Protein	31.9	32.7	32.1	35.0	0.37	<0.01	0.03	0.02
Fat	38.7	41.8	42.8	47.4	2.45	0.05	0.78	0.63
Lactose	47.5	48.5	48.2	49.7	0.49	0.03	0.64	0.22
Solid non-fat	87.1	88.7	88.2	92.6	0.96	<0.01	0.2	0.16
Milk energy, MJ/kg	3.0	3.2	3.2	3.5	0.11	0.03	0.61	0.44
SCC, ×10^3^ cells/mL	380	276	267	557	278	0.14	0.44	0.71
Feed efficiency	1.1	1.1	1.0	1.0	0.22	0.75	0.99	0.60

^1^ Fat and protein-corrected milk was calculated by Gerber et al. [17]. FPCM (kg/d) = (0.337 + 0.116 × fat% + 0.06 × protein%) × milk yield (kg/d). Energy-corrected milk (ECM, kg/d) = (milk yield (kg/d) × milk energy (MJ/kg))/3.1 and milk energy (MJ/kg) = (0.0384 × fat) + (0.0223 × protein) + (0.0199 × lactose) − 0.108, as described by Cabezas-Garcia et al. [18]. SCC = somatic cell count. Feed efficiency = FPCM yield (kg/d)/DMI (kg/d). ^2^ Probability of significant linear (L), quadratic (Q), and cubic (C) effects of cassava chip levels.

**Table 6 animals-13-02296-t006:** Daily respiratory gas consumption of oxygen, production of carbon dioxide, and enteric methane emission from lactating dairy cows (*n* = 4 per treatment) fed a fermented total mixed ration at different levels of cassava chip substitution.

Items ^1^	Cassava Chip Substitution (%)				SEM	*p*-Value ^2^		
	0	12	24	36		L	Q	C
Respiratory gas								
O_2_ consumption, L/d	3705	3778	3549	3904	62.94	0.24	0.07	0.02
CO_2_ production, L/d	4189	4399	4205	4594	55.27	<0.01	0.16	<0.01
RQ	1.13	1.16	1.18	1.19	0.01	<0.01	0.09	0.97
Enteric methane emission								
L/d	304.8	342.1	355.0	351.8	13.09	0.04	0.17	0.89
L/kg DMI	24.2	27.9	24.4	23.7	1.79	0.55	0.27	0.26
L/kg FPCM	23.6	26.3	29.4	25.5	1.95	0.35	0.14	0.42
MJ/100 MJ GEI	6.4	7.7	7.6	6.9	0.18	0.13	<0.01	0.37

^1^ Respiratory quotient = CO_2_ production/O_2_ consumption. DMI = dry matter intake. Fat- and protein-corrected milk was calculated by Gerber et al. [17]. FPCM (kg/d) = (0.337 + 0.116 × fat% + 0.06 × protein%) × milk yield (kg/d). GEI = gross energy intake. ^2^ Probability of significant linear (L), quadratic (Q), and cubic (C) effects of cassava chip levels.

**Table 7 animals-13-02296-t007:** Energy partitioning, intake, content of diet, and utilization in lactating dairy cows (*n* = 4 per treatment) fed a fermented total mixed ration at different levels of cassava chip substitution.

Items ^1^	Cassava Chip Substitution (%)				SEM	*p*-Value ^2^		
	0	12	24	36		L	Q	C
Energy partition, kJ/kg BW^0.75^								
GE intake	1887	1819	1834	1972	57.35	0.33	0.12	0.89
Fecal excretion	603	531	518	530	18.64	0.03	0.06	0.70
Urine excretion	60.8	50.9	44.8	39.1	3.35	<0.01	0.56	0.83
Methane emission	120	134	137	135	4.50	0.06	0.12	0.83
Heat production	782	802	751	820	7.49	0.11	0.02	<0.01
Milk energy	337	352	342	365	23.23	0.50	0.88	0.61
Energy balance	−15.3	−51.0	41.2	83.5	51.04	0.14	0.47	0.47
E_l(0)_	334	309	385	448	43.02	0.07	0.35	0.58
Energy intake, kJ/kg BW^0.75^								
DE	1284	1288	1316	1442	49.48	0.06	0.26	0.75
ME	1104	1103	1135	1268	48.01	0.05	0.21	0.75
NE_L_	844	837	822	959	110.23	0.53	0.54	0.76
Energy content, MJ/kg of DM								
DE	12.4	12.9	13.1	13.1	0.21	0.05	0.29	0.99
ME	10.6	11.0	11.3	11.5	0.23	0.03	0.67	0.90
NE_L_	8.0	8.4	8.0	8.7	1.02	0.75	0.87	0.69
Energy utilization								
DE/GE	0.68	0.71	0.73	0.74	0.01	<0.01	0.34	0.87
ME/GE	0.58	0.61	0.63	0.65	0.01	<0.01	0.78	0.80
ME/DE	0.85	0.85	0.86	0.88	0.01	0.03	0.25	0.74

^1^ E_l(0)_ = milk energy output adjusted to zero energy balance. DE = digestible energy. GE = gross energy. ME = metabolizable energy. NE_L_ = net energy for lactation. ^2^ Probability of significant linear (L), quadratic (Q), and cubic (C) effects of cassava chip levels.

**Table 8 animals-13-02296-t008:** Regression method used for estimating energy values in cassava chips.

Item ^1^	Regression Equations ^2^	RMSE ^3^	R^2^	*p*-Value	Slope		Intercept	
					SEM	*p*-Value	SEM	*p*-Value
DE, MJ/kg DM	*Y* = 12.42*X* + 0.002	0.17	0.99	<0.001	0.31	<0.001	0.07	0.97
ME, MJ/kg DM	*Y* = 10.63*X* + 0.002	0.20	0.98	<0.001	0.38	<0.001	0.09	0.98
NE_L_, MJ/kg DM	*Y* = 8.04*X* + 0	0.65	0.76	<0.001	1.21	<0.001	0.27	0.99

^1^ DE = digestible energy. ME = metabolizable energy. NE_L_ = net energy for lactation. DM = dry matter. ^2^
*Y* is ingredient-associated digestible energy, metabolizable energy, and net energy for lactation in megajoules. *X* is test ingredient intake in kilograms (DM basis); the intercept is in megajoules, and the slopes are in megajoules per kilograms DM. ^3^ RMSE = root–mean square error.

**Table 9 animals-13-02296-t009:** Analyzed energy values of cassava chip feedstuff determined by substitution methods when compared with the regression method.

Item ^1^	Comparative Method				SEM	*p*-Value
	Substitution	Substitution	Substitution	Regression		
	12%	24%	36%			
DE, MJ/kg DM	12.43	12.44	12.43	12.42	0.39	1.00
ME, MJ/kg DM	10.63	10.65	10.65	10.63	0.48	1.00
NE_L_, MJ/kg DM	8.01	8.03	8.05	8.04	1.54	1.00

^1^ DE = digestible energy, ME = metabolizable energy, NE_L_ = net energy for lactation, DM = dry matter.

## Data Availability

Not applicable.

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
