# Peer review of "Net Energy Value of a Cassava Chip Ration for Lactation in Holstein–Friesian Crossbred Dairy Cattle Estimated by Indirect Calorimetry"

_animals, 2023, doi:10.3390/ani13142296_

Round 1
Reviewer 1 Report
This manuscript describes an experiment to determine the energy value of cassava chips. There are no major concerns with this study.
Overall, English grammar is good, but over use of commas. Please revise throughout the manuscript.
Specific comments:
L22 - change to 'digestibility, energy balance, milk yield'
L23 - change to 'without affecting enteric'
L24 - change to '8.03 MJ/kg DM.'
L25 - change to digestibility, and energy'
L38 - change to 'MJ/kg DM.'
L67 - change to 'University research station'
L92 - change to 'ingredients were mixed'
L93-94 - this sentence does not belong in this section. It belongs in the Calculations section.
L116 - change to 'animals were relocated'
L117 - change to 'experimental period.'
L132 - change to 'in the hood and had'
L145 - what is the unit for the value 7.5?
L154 - change to 'feces were analyzed'
L182-183 - Equation 3 should preceede Equation 2 because the result of Equation 3 is used in Equation 2.
L228 - change to 'The chemical composition and energy'
Table 2 - change title to read 'composition, energy'. A couple of changes in the nutrient profile of the diet could have affected the results. Decreasing CP likely decreased rumen degradable protein negatively affecting NFC digestibility and the NEl. Also lower CP would result in cows mobilizing amino acids from body tissues, which would affect NEl. The increasing NFC in the diet could be causing subacute rumen acidosis which negatively affects milk yield, thus affecting NEl.
Table 3 - %BW and g/kg BW are providing the same information. I dont think you need both.
Table 6 - The RQ values for 0% and 12% are greater than 1. This seems high for cattle in negative energy balance that would be mobilizing and catabolizing fat for energy. Are you sure these are correct?
L296 - change to 'tendencies for linear'
L297 - change to 'DE tended to increase (p = 0.06) and ME'
L298 - change to 'diets increased'
Table 7 - reword title to 'partitioning, intake'
Figure 1 - most places you use 'different substitutions of cassava chips', but here you use proportions. Please be consistent.
L341 - change to 'slopes are in'
L355 - change to 'improvement, suggesting'
L361 - change to 'resulted in increased'
L376 - change to 'that the milk fat'
L383 - change to 'effect on the methane'
L388 - change to 'were like Mickayla [25] who reported methane'
L399 - change to 'consistent with previous'
L446 - I disagree that diet has no effect on kl values. If this were true, then all feedstuffs would have the same NEl value.
L453 - here you indicate that genetic selection has improved efficiency of ME use (kl), but in L446 you indicate that animal has no impact on kl. These contradict each other. Please correct this.
L474 - what does the value 11.8% mean? 11.8% of what?
L485-487 - the word 'values' should be the singular 'value'
Overall english grammar is good. Most of my comments are correcting minor grammatical problems. Also, there is over use of commas.
Author Response
Point-by-point Responses to Reviewer Comments
Response to Reviewer 1 Comments
Comments and Suggestions for Authors
This manuscript describes an experiment to determine the energy value of cassava chips. There are no major concerns with this study.
Overall, English grammar is good, but over use of commas. Please revise throughout the manuscript.
Specific comments:
Point 1:
L22 - change to 'digestibility, energy balance, milk yield'
L23 - change to 'without affecting enteric'
L24 - change to '8.03 MJ/kg DM.'
L25 - change to digestibility, and energy'
L38 - change to 'MJ/kg DM.'
L67 - change to 'University research station'
L92 - change to 'ingredients were mixed'
Response 1: Thank you for your recommendation. We made a correction accordingly.
Point 2:
L93-94 - this sentence does not belong in this section. It belongs in the Calculations section.
Response 2: Thank you for your suggestion. The text now moves to the calculation section (2.5).
Point 3:
L116 - change to 'animals were relocated'
L117 - change to 'experimental period.'
L132 - change to 'in the hood and had'
L145 - what is the unit for the value 7.5?
L154 - change to 'feces were analyzed'
Response 3: We made a correction accordingly.
Point 4:
L182-183 - Equation 3 should preceede Equation 2 because the result of Equation 3 is used in Equation 2.
Response 4: We made a correction accordingly.
Point 5:
L228 - change to 'The chemical composition and energy'
Table 2 - change title to read 'composition, energy'. A couple of changes in the nutrient profile of the diet could have affected the results. Decreasing CP likely decreased rumen degradable protein negatively affecting NFC digestibility and the NEl. Also lower CP would result in cows mobilizing amino acids from body tissues, which would affect NEl. The increasing NFC in the diet could be causing subacute rumen acidosis which negatively affects milk yield, thus affecting NEl.
Response 5: Thank you for your observation. To improve clarity, we rewrote this passage and added details for clarity in the Results section (L230-233) as follows. “The OM, NDF, and ADF content in dietary treatments was similar among diets. The CP and EE were decreased, but NFC and ME content increased substantially when increasing cassava in the diets.”
Also, in the Discussion section (L368-370) as follows. “The DM, OM, and NFC intakes increased when increasing the level of cassava chips in the diet, which resulted in increased digestibility of DM, OM, and NFC without any adverse effect on fiber digestion and rumen fermentation.”
Point 6:
Table 3 - %BW and g/kg BW are providing the same information. I dont think you need both.
Response 1: Thank you for this clarification. We have removed data on feed intake (g/kg BW) for clarity.
Point 7:
Table 6 - The RQ values for 0% and 12% are greater than 1. This seems high for cattle in negative energy balance that would be mobilizing and catabolizing fat for energy. Are you sure these are correct?
Response 7: Thank you for your observation. To improve clarity, we rewrote this passage and added details for clarity in Discussion section L415-424 as follows.
“The respiratory quotient (RQ) is the ratio of carbon dioxide production to oxygen consumption [30]. The RQ value greater than 1 can be used as an approximate indicator of carbohydrates being utilized for fat synthesis that is occurring in the animal [31]. The RQ values increased when increasing the level of cassava chips in the diet; the average value of RQ in this study (ranging from 1.13 to 1.19) was close to the RQ values reported for dairy cows using the respiration chamber technique, which ranged from 1.01 to 1.17 [30,32]. Our result indicated that NFC intake and digestibility increase associated with the increasing RQ, milk fat content and yield, confirming a higher milk fatty acid synthesis from carbohydrate utilization when increasing the level of cassava chips in the diet.”
Point 8:
L296 - change to 'tendencies for linear'
L297 - change to 'DE tended to increase (p = 0.06) and ME'
L298 - change to 'diets increased'
Table 7 - reword title to 'partitioning, intake'
Response 8: We made a correction accordingly.
Point 9:
Figure 1 - most places you use 'different substitutions of cassava chips', but here you use proportions. Please be consistent.
Response 9: Thank you for your advice. We have reworded it to “different substitutions of cassava chips”.
Point 10:
L341 - change to 'slopes are in'
L355 - change to 'improvement, suggesting'
L361 - change to 'resulted in increased'
L376 - change to 'that the milk fat'
L383 - change to 'effect on the methane'
L388 - change to 'were like Mickayla [25] who reported methane'
L399 - change to 'consistent with previous'
Response 10: We made a correction accordingly.
Point 11:
L446 - I disagree that diet has no effect on kl values. If this were true, then all feedstuffs would have the same NEl value.
L453 - here you indicate that genetic selection has improved efficiency of ME use (kl), but in L446 you indicate that animal has no impact on kl. These contradict each other. Please correct this.
Response 12: Thank you for your advice. We agree on our missing and revised as follows.
“In the present study, the mean kl value was 0.59, which was almost identical to the reported value of 0.53 of Bos taurus x Bos indicus dairy cows under tropical conditions by Oliveira [34] but less than the value of Bos taurus [31,36,39]. Oliveira [34] reported that the lower kl value of the crossbred dairy cows in tropical conditions could be due to the genotypes, diets, and differences in homeostatic regulation. These discrepancies between the estimated and observed maintenance expenditures and efficiency of ME use for lactation reflect and highlight the progress of modern dairy cattle breeding, which focuses on improvements through genetic selection [35].”
Point 13:
L474 - what does the value 11.8% mean? 11.8% of what?
Response 13: Thank you for your observation. To improve clarity, we rewrote it as follows.
“In addition, the NEL content (8.04 MJ NEL/kg DM) of cassava chips in this study was found to be 9.8% higher than the value of 7.32 MJ NEL/kg DM reported by WTSR [8].”
Point 14:
L485-487 - the word 'values' should be the singular 'value'
Response 14: We made a correction accordingly.
Point 15:
Comments on the Quality of English Language
Overall English grammar is good. Most of my comments are correcting minor grammatical problems. Also, there is over use of commas.
Response 15: Thank you for your helpful feedback.

Reviewer 2 Report
Very well written manuscript.
My biggest concern is the choice to replace the entire diet with Cassava chip product instead of part of the concentrate. Also, utilization of cassava pulp as an ingredient seems to be a confounding aspect to the ingredients selected for inclusion in the diet.
The research is somewhat under powered with the use of only a single 4x4 Latin square. A replicated Latin square would be advisable in this research.
the use of Contrasts along with mean separation using Duncans is unnecessary and the use of only the contrasts provides more clarity to the presentation of the data.
Over all a well designed study and a well written manuscript.
Author Response
Response to Reviewer 2 Comments
Comments and Suggestions for Authors
Very well written manuscript.
Point 1: My biggest concern is the choice to replace the entire diet with Cassava chip product instead of part of the concentrate. Also, utilization of cassava pulp as an ingredient seems to be a confounding aspect to the ingredients selected for inclusion in the diet.
Response 1: Thank you for your observation. To improve clarity, we add the limitation of this experiment in the discussion section (L463) as follows.
“The limitation of this experiment was also due to the increase in the cassava levels in the diets by a single-point substitution or multiple-point substitution method required for evaluating the energy content of individual feedstuffs. The method involves different substitution rates of a test feed (cassava chips) into each energy feed source in the basal diets. Therefore, the effects observed are not only due to the cassava increase but also to diet composition change.”
References
A single substitution or multiple-substitution method is required for evaluating the energy content of individual feedstuffs. This method involves different substitution rates of a test feed (cassava chips) into each energy feed source (not only part of the concentrate) in the basal diets. In this study, the 4 dietary treatments consisted of 1 basal diet that contained 0% cassava chip, and 3 test diets. In the basal diet, 6 feedstuffs were rice straw, cassava pulp, wet brewery waste, rice bran, palm kernel cake, and soybean meal were the sources of energy. In the 3 test diets, 6 energy feedstuffs were partly substituted by cassava chips at 12%, 24%, and 36% so that the ratio of 6 energy feedstuffs was equal in the basal diet and these 3 test diets (Table 1).
Substitution and multiple-substitution methods were used worldwide to evaluate the energy content of individual feedstuffs [1-5].
- Wei, M.; Cui, Z.H.; Li, J.W.; Yan, P.S. Estimation of metabolisable energy and net energy of rice straw and wheat straw for beef cattle by indirect calorimetry. Anim. Nutr. 2018, 72, 275-289. DOI:10.1080/1745039X.2018.1482076.
- Adeola, O; Ileleji, K.E. Comparison of two diet types in the determination of metabolizable energy content of corn distillers dried grains with solubles for broiler chickens by the regression method. Sci. 2009, 88, 579-585. DOI:10.3382/ps.2008-00187.
- Bolarinwa, O.A.; Adeola, O. Energy value of wheat, barley, and wheat dried distillers grains with solubles for broiler chickens determined using the regression method. Sci. 2012, 91, 1928–1935. DOI:10.3382/ps.2012-02261.
- Osunbami, O.T.; Aderibigbe, A.S.; Adeola, O. Energy value of dry fat and stabilised rice bran for broiler chickens. Poult. Sci. 2021, 62, 835–839. DOI:10.1080/00071668.2021.1940863.
- Osunbami, O.T.; Adeola, O. Regression method-derived digestible and metabolizable energy concentrations of partially defatted black soldier fly larvae meal for broiler chickens and pigs. Sci. 2022, 264, 105042. DOI:10.1016/j.livsci.2022.105042.
Point 2: The research is somewhat under powered with the use of only a single 4x4 Latin square. A replicated Latin square would be advisable in this research.
Response 2: Thank you for your recommendation. We have added limitations of the study using a 4x4 Latin square for clarity in the Discussion section (L491-495) and Conclusion (L507-509) as follows.
“Although a 4x4 Latin square design accounted for cow and period effects and allowed for measurements of animal responses to changes in dietary treatment, however, the relatively short (21 d) time of the periods in the Latin square design and the limited number of cows per treatment are limitations of this study. In particular, the milk production performance results need to be confirmed over a longer feeding experimental period and a greater number of animals.
Long-term feeding trials or on-farm research is needed for the development of a practical and economical dairy zebu crossbred fed cassava-based diets feeding system.”
Point 3: The use of Contrasts along with mean separation using Duncans is unnecessary and the use of only the contrasts provides more clarity to the presentation of the data.
Response 3: Thank you for this clarification. We made a correction accordingly.
Overall a well designed study and a well written manuscript.
